# Effect of a Third Dose of SARS-CoV-2 mRNA BNT162b2 Vaccine on Humoral and Cellular Responses and Serum Anti-HLA Antibodies in Kidney Transplant Recipients

**DOI:** 10.3390/vaccines10060921

**Published:** 2022-06-09

**Authors:** Irene Cassaniti, Marilena Gregorini, Federica Bergami, Francesca Arena, Josè Camilla Sammartino, Elena Percivalle, Ehsan Soleymaninejadian, Massimo Abelli, Elena Ticozzelli, Angela Nocco, Francesca Minero, Eleonora Francesca Pattonieri, Daniele Lilleri, Teresa Rampino, Fausto Baldanti

**Affiliations:** 1Microbiology and Virology Department, Fondazione IRCCS Policlinico San Matteo, 27100 Pavia, Italy; i.cassaniti@smatteo.pv.it (I.C.); federica.bergami01@universitadipavia.com (F.B.); francesca.arena01@universitadipavia.it (F.A.); jose.sammartino@iusspavia.it (J.C.S.); e.percivalle@smatteo.pv.it (E.P.); ehsan.soleymaninejadian@gmail.com (E.S.); d.lilleri@smatteo.pv.it (D.L.); fausto.baldanti@unipv.it (F.B.); 2Department of Internal Medicine and Therapeutics, University of Pavia, 27100 Pavia, Italy; 3Unit of Nephrology, Dialysis, Transplantation, Fondazione IRCCS Policlinico San Matteo, 27100 Pavia, Italy; francesca.minero01@universitadipavia.it (F.M.); ef.pattonieri@gmail.com (E.F.P.); t.rampino@smatteo.pv.it (T.R.); 4Transplant Unit, Fondazione IRCCS Policlinico San Matteo, 27100 Pavia, Italy; m.abelli@smatteo.pv.it; 5Unit of General Surgery, Fondazione IRCCS Policlinico San Matteo, 27100 Pavia, Italy; e.ticozzelli@smatteo.pv.it; 6Laboratory of Transplant Immunology, Fondazione IRCCS Ca’ Granda, Ospedale Maggiore Policlinico, 20122 Milan, Italy; angela.nocco@policlinico.mi.it; 7Department of Clinical, Surgical, Diagnostic and Pediatric Sciences, University of Pavia, 27100 Pavia, Italy

**Keywords:** transplanted patients, SARS-CoV-2, BNT162b2 vaccine, third dose, kidney, DSA, anti-HLA antibodies

## Abstract

The severe acute respiratory syndrome coronavirus 2 (SARS-CoV-2) pandemic has severely impacted on public health, mainly on immunosuppressed patients, including solid organ transplant recipients. Vaccination represents a valuable tool for the prevention of severe SARS-CoV-2 infection, and the immunogenicity of mRNA vaccines has been evaluated in transplanted patients. In this study, we investigated the role of a third dose of the BNT162b2 vaccine in a cohort of kidney transplant recipients, analyzing both humoral and cell-mediated responses. We observed an increased immune response after the third dose of the vaccine, especially in terms of Spike-specific T cell response. The level of seroconversion remained lower than 50% even after the administration of the third dose. Mycophenolate treatment, steroid administration and age seemed to be associated with a poor immune response. In our cohort, 11/45 patients experienced a SARS-CoV-2 infection after the third vaccine dose. HLA antibodies appearance was recorded in 7 out 45 (15.5%) patients, but none of the patients developed acute renal rejection. Further studies for the evaluation of long-term immune responses are still ongoing, and the impact of a fourth dose of the vaccine will be evaluated.

## 1. Introduction

The current pandemic caused by the severe acute respiratory syndrome coronavirus-2 (SARS-CoV-2) is representing one of the major hurdles in solid organ transplant recipients (SOTRs), with a mortality ranging from 18% to 30% [1]. On the other hand, vaccination seems to be effective in preventing COVID-19 in SOTRs. Kamar and colleagues demonstrated that COVID-19 vaccination in SOTRs induced an immune response, even if the levels of serum antibodies were lower than those measured in healthy controls [2,3]. After each dose of BNT162b2 vaccine (Pfizer-BioNTech), serum antibodies against SARS-CoV-2 increased significantly from a signal-to-cut-off ratio of 36 ± 12 before the third dose to a value of 2676 ± 350 1 month after the third dose [4]. The averages of detectable antibodies against the spike protein after the first and second doses of vaccination were less than 15% [5] and 50% [6], respectively, in kidney transplant patients. As a result of the low efficacy of two doses of vaccination in SORTs, the third dose of vaccination in these patients became a necessity. Additionally, the first period after transplant is associated with the highest level of immunosuppression, and vaccinations are not recommended due to the low rate of responsiveness [7]. Then, six months post-transplant, a vaccination plan guided by a reduction in immunosuppression can lead to a better response to antivirals and vaccines.

A crucial role of the T cell-mediated response elicited by SARS-CoV-2 vaccination may be hypothesized. Parallel analyses of the serum IgG level and T cell response underlined the importance of cellular immunity in immunocompromised patients vaccinated against SARS-CoV-2. Indeed, despite the absence of antibody response, most of the immune responses were associated with Spike-specific T cells [2,8]. Interestingly, though CD8 T responses are dominant in the SARS-CoV-2-infected patients with mild symptoms, the overall response of T cells is higher in patients with severe symptoms [9]. Furthermore, the CD8 T cells seem to play a pivotal role in protecting vaccinated people in the early days after vaccination. This stage precedes the reaction of antibodies or even CD4 T cells with the viral epitopes [10]. In addition to T cells, binding antibodies such as IgG and IgA displayed an important role in the early stage of vaccination—less than 10 days from vaccine injection—in comparison to neutralizing antibodies or receptor-blocking antibodies [11]. An important open issue is whether vaccination may represent a non-specific trigger factor for developing de novo donor-specific antibodies (DSA) or anti-HLA antibodies (human leukocytes antigens) that are associated with renal rejection. 

Therefore, in this study, we focused on the immune responses in kidney transplanted patients after the third dose of Pfizer-BioNTech vaccine, especially on T cells responses in the early stage of vaccination. Secondary aims were the evaluation of the influence of vaccination on anti-HLA and anti-DSA antibodies appearance as well as rejection occurrence. 

## 2. Methods

### 2.1. Patients and Samples

BNT162b2-vaccinated kidney transplant recipients (KTRs) were prospectively enrolled at the time of first-dose vaccination (April 2021), and the immune response elicited by vaccination was analyzed according to the following time-points: (i) baseline (before vaccination; T0); (ii) 42 days (T1) (three weeks after the second dose); (iii) six months after the second dose and before the administration of the third dose (T2); (iv) 21 days after the third dose (T3) and (v) 3 months after the third dose (T4) (Figure 1). The study follow-up ended in March 2022.

Based on the total anti-Spike response observed at the time of enrolment or on the documented history of positive nasal swabs for SARS-CoV-2, only COVID-19-naïve (negative for SARS-CoV-2 RNA and/or negative for anti-Spike IgG) patients were included in further analyses. The study was conducted according to the guidelines of the Declaration of Helsinki and approved by the Ethics Committee “Comitato Etico Pavia” (P-20210000232) on 10 February 2021. Patients gave informed consent for their data to be anonymously utilized for a scientific scope according to the policy of the protocol.

### 2.2. SARS-CoV-2 Humoral Response

A chemiluminescent assay (Liaison SARS-CoV-2 trimeric, Diasorin) was used for Spike IgG quantification. Values higher than 33.8 BAU/mL were considered positive. SARS-CoV-2 neutralizing antibody titer (NT Abs) was determined as previously reported [12,13]. Briefly, 50 µL of serum in serial fourfold dilution, was placed in two wells of a flat-bottom tissue culture microtiter plate (COSTAR, Corning Incorporated, New York, NY, USA). The same volume of 100 TCID50 of SARS-CoV-2 strain was added, and the plates were incubated at 33 °C in 5% CO_2_. After 1 h of incubation at 33 °C and 5% CO_2_, VERO E6 cells were added to each well. After further 72 h of incubation at 33 °C and 5% CO_2_, the plates were stained with Gram’s crystal violet solution (Merck KGaA, Damstadt, Germany) plus 5% formaldehyde 40% m/v (Carlo ErbaSpA, Arese, Italy) for 30 min. The microtiter plates were then washed under running water. The wells were scored to evaluate the degree of cytopathic effect (CPE) compared to the virus control. A blue staining of the wells indicated the presence of NT Abs. The neutralizing titer was the maximum dilution showing the reduction of 90% of CPE. All the experiments were performed in a BSL3 facility. Values higher or equal to 1:10 serum titer were considered positive, according to our protocol.

### 2.3. SARS-CoV-2 Spike-Specific T Cell Response

Peripheral blood mononuclear cells (PBMC) were isolated from heparin-treated blood by standard density gradient centrifugation and used as described below. Briefly, membrane-bottomed 96-well plates were coated with an anti-IFN-γ monoclonal capture antibody (Human IFN-γ ELISpot kit, Diaclone, Pantec, Kradolf-Schönenberg, Switzerland) and kept at 4 °C overnight. Then, PBMC (2 × 10^5^/100 μL culture medium per well) were stimulated in duplicate for 24 h with peptide pools (15 mers, overlapping by 10 amino acids, Pepscan, Lelystad, The Netherlands) representative of the whole spike protein (S) at the final concentration of 0.25 µg/mL. Phytohemagglutinin (PHA; 5 µg/mL) was used as a positive control, and medium alone was used as a negative control. The enzyme-linked immunospot assay (ELISpot) was performed according to our previous protocol [14]. Responses ≥10 IFN-γ-producing cells/10^6^ PBMC were considered positive based on background results obtained with the negative control (mean SFC + 2SD).

### 2.4. Anti-HLA Antibodies Determination

The presence of anti-HLA antibodies (DSA and not DSA) was tested using the Luminex technology. Serum samples from the recipients collected before the first dose and after the third dose of BNT 162 b2 vaccine were analyzed for class I and class II IgG HLA antibodies using the commercially available LABScreen Single Antigen Beads Class I and Class II Assay Kit (One Lambda, West Hills, CA, USA). The procedure was performed as previously reported [15], and the data were analyzed on a LABScan200 flow analyzer (One Lambda). The results were interpreted using Mean Fluorescence Intensity (MFI) values. All samples were considered positive if the MFI value was >1000.

### 2.5. Clinical and Therapeutic Variables

Patients enrolled were on regular follow-up at the outpatient kidney transplant unit. Biochemical measurements were performed according to the Center’s policy. Clinical data and results of laboratory and immunosuppressive drugs were recorded and analyzed. The immunosuppressive induction treatment consisted of anti-thymocyte globulins (ATG) or Basiliximab, and the maintenance regimen included calcineurin inhibitors (such as cyclosporine or tacrolimus), mTOR inhibitors (mTORi) (such as sirolimus or everolimus), antiproliferative drugs (such as mycophenolate mofetil/mycophenolic acid or azathioprine) and steroids, combined in triple, double or monotherapy.

### 2.6. COVID19 Diagnosis

Patients with fever or symptoms suggestive of COVID-19 were screened for SARS-CoV-2 infection through the identification of virus-unique RNA sequences by real-time reverse transcription polymerase chain reaction (RT-PCR) on nasopharyngeal or oropharyngeal swab.

### 2.7. Statistical Analysis

The frequency and percentage of subjects positive for total IgG, SARS-CoV-2 NT Abs and S-ELISpot (Spike-specific T cell response) were determined, and comparisons between groups were made by Fisher’s exact test. Quantitative data are presented as median and interquartile range (IQR), data were log-transformed, and comparisons were made using the ANOVA test with Turkey correction. The Spearman test was used for correlation analyses. A multiple linear regression analysis was adopted to identify independent predictors of the immune response to the vaccine. Immune parameters were log-transformed for the analysis.

All the assays were two-tailed, and a *p* value < 0.05 was considered significant. GraphPad Prism 8.3.0 (GraphPad Software Inc., La Jolla, CA, USA) was used for all the analyses.

## 3. Results

### 3.1. Clinical and Demographic Characteristics

The mean age of the patients enrolled was 52.6 years, 23 patients were males. The median transplant age was 51 months. Induction therapy was performed with Thymoglobulin in 15.5% (7/45) and with Basiliximab in 84.4% (38/45) of the patients. All the patients received immunosuppressive drugs at the time of vaccination. In 29 (64.4%) patients, triple immunosuppressive therapy was administrated (steroid, calcineurin inhibitors and mTORi or antimetabolites), 15 (33.3%) patients received two immunosuppressive drugs (calcineurin inhibitors and mTORi or antimetabolites or steroid), and only 1 (2.2%) patient received one immunosuppressive drug (steroid). In Table 1, the demographic and clinical characteristics of the enrolled patient are summarized.

### 3.2. Immunogenicity of BNT162b2 Vaccination

The patients studied were seronegative at the time of administration of the first dose (total IgG-negative); An S-ELISpot response was detected in 17/42 (40.5%) subjects. After two doses of BNT162b2 vaccine, IgG prevalence was 25%, while positive SARS-CoV-2 NT Abs and S-ELISpot levels were detected in 35% and 47.6% of patients, respectively. Six months after vaccination, the prevalence was 45%, 38% and 59.5% for total IgG, SARS-CoV-2 NT Abs and S-ELISpot. One month after the third dose, the prevalence for a positive humoral response was 53% and 60%, in terms of total IgG and SARS-CoV-2 NT Abs, respectively. On the other hand, the rate of responders was 75.6% when the Spike-specific T cell response was measured (Figure 2). 

### 3.3. Longitudinal Monitoring of SARS-CoV-2 Humoral Response and Spike-Specific T Cell Response Elicited by Vaccination

The median levels of total anti-Spike IgG were 4.8 IQR 4.8–85.3 BAU/mL and 6.9 IQR 4.8–88.4 BAU/mL after two doses of BNT162b2 vaccine and after six months from the last dose, respectively. We did not observe a significant decrease of the IgG level between the two time points, since the large majority of patients were still negative at T2. Three weeks after the third dose, the level of response reached the maximum median level of 52.5 IQR 4.8–1178 BAU/mL and decreased to 41.4 IQR 6.5–650 BAU/mL at three months after the third dose (Figure 3A). Additionally, SARS-CoV-2 NT Abs levels were <1:10 IQR < 1:10–1:20 and <1:10 IQR < 1:10–1:10 three weeks and six months after the second dose, respectively. The level of NT Abs reached 1:20 IQR 1:5–1:160 and 1:10 IQR < 1:10–1:40 three weeks and three months after the third dose, respectively (Figure 3B).

In parallel, the T cell response against the Spike antigen was measured at the same time points. The median level of response three weeks after the second dose was 5.0 IQR 1.0–10.5 Spike-specific IFNγ-producing T cells and reached 7.5 IQR 1.0–20.0 Spike-specific IFNγ-producing T cells after six months. A sustained increase was observed after the third dose (median 25 IQR 7.5–85.0 Spike-specific IFNγ-producing T cells; *p* = 0.013). The median level of response was 20 IQR 5.0–70.0 Spike-specific IFNγ-producing T cells when measured three months after the third dose (Figure 3C). 

In order to analyze the potential impact of baseline Spike-specific T cell response in vaccine immunogenicity, we compared the responses between subjects with negative and positive Spike-specific T cell responses at baseline. However, no significant difference was observed (Appendix A).

### 3.4. Demographic and Clinical Parameters Correlation

Demographic and clinical parameters including age, months from transplantation and immunosuppressive regimens were analyzed in univariate regression analysis, and the impact of each variable on BNT162b2-elicited immune response was evaluated. Overall, we observed that patient’s age was inversely correlated with Spike-specific T cell response measured three weeks after the third dose [r = −0.434; IC95 −0.65–(−0.14); *p* = 0.0046]; indeed, the median age of S-ELISpot responders and non-responders was 51 [IQR 43–58] and 61 [IQR 54–65] years, respectively (*p* = 0.0028). Otherwise, no association between months from transplantation and BNT162b2 vaccine response was observed, since the median level of months after transplantation was similar between responder and non-responder subjects, in terms of both humoral (*p* = 0.7042) and T cell-mediated response (*p* = 0.5474). An association between mycophenolate treatment and level of trimeric anti-Spike IgG but not NT Abs and Spike-specific T cell response measured three weeks after the third dose was detected. In detail, a total of 16/19 (84.2%) subjects with no detectable anti-Spike IgG level were treated with mycophenolate (Figure 4). The impact of the steroid treatment was also evaluated and, even if no differences were reported in terms of cell-mediated response and total IgG level, a significant difference between steroid-treated and -untreated patients was observed when SARS-CoV-2 NT Abs were evaluated (Figure 5).

In a multivariate linear regression model, we found that the use of mycophenolate was significantly associated with a lower anti-S trimeric IgG antibody level, while the association between use of steroid and a low antibody level was close to significance. Conversely, age and use of steroid were significantly associated with a lower T cell response. The association between age or use of mycophenolate and a lower NT Abs titer was close to significance (Table 2).

### 3.5. Anti-HLA Antibodies

HLA antibodies appearance was recorded in 7 out 45 (15.5%) patients after the end of the vaccination cycle. Two patients (4.4%) developed de novo DSA antibodies, and one patient (2.2%) showed an increase in the titer of pre-existing DSA antibodies. De novo HLA antibodies (non-DSA) appeared in 4fourpatients (8.8%). It is noteworthy that only one patient with HLA antibodies had never developed neutralizing antibodies. No rejection episodes were recorded during this follow-up period.

### 3.6. COVID-19 Cases in Vaccinated Transplanted Patients

Overall, 11/45 (24.4%) patients reported SARS-CoV-2 infection after administration of the third dose (median days after the third dose, 106; range, 92–117). Symptoms and treatment, as well as SARS-CoV-2 immune parameters measured before SARS-CoV-2 infection, are reported in Table 3. Of note, two of these patients reported a previous SARS-CoV-2 infection before vaccination. No differences in terms of median levels of SARS-CoV-2 immune response elicited by BNT162b2 vaccination were reported between infected and uninfected subjects after the third vaccination dose. In this cohort, no patient reported SARS-CoV-2 infection between the second and the third dose of vaccine administration. 

## 4. Discussion

It is well known that SOTRs as well as hemodialyzed patients are at high risk of COVID-19 severe infection, and vaccination represents a valuable tool for the prevention of SARS-CoV-2 infection [16,17,18]. In this prospective longitudinal study, we evaluated the immunogenicity of three doses of BNT162b2 vaccine in a small cohort of KTRs, analyzing both humoral and cell-mediated responses. 

The rate of so-called “humoral responders” (responders for total IgG and/or SARS-CoV-2 NT Abs) was about 50%. On the other hand, we reported a detectable baseline T cell response in about 40% of the enrolled subjects, suggesting that a cross-reactive T cell response elicited by previous human common coronaviruses (HCoVs) might be present, as previously reported [14]. Thus, the third vaccination was able to elicit a de-novo T cell response in the other 35% of the subjects, leading to an increase also in terms of median response in the overall positive subjects. 

Age is related to the T cell-mediated response after the third dose, since S-ELISpot responders were younger than non-responders. In our study, age seems not to be related to the rate of humoral seroconversion. On the other hand, Del Bello and colleagues reported that younger patients showed a higher seroconversion rate [19]. There are several factors that may affect the overall immune response elicited by vaccination, including the amount of immunosuppressive drugs taken, vintage transplant and the type of vaccine, since better responses were observed against mRNA-1273 (Moderna, Cambridge, MA, USA).

The impact of immunosuppressive drugs on SARS-CoV-2 vaccine responsiveness represents a crucial point. Looking at the immunosuppressive regimens, the administration of mycophenolate is associated with a reduced immunogenicity, as observed in our previous study [2] and many other studies [19,20,21]. Additionally, we observed an impact of steroid treatment on immunogenicity after the third dose, especially in T cell response. Other studies reported a poor immune response in patients treated with belatacept [21,22,23]. So far, in vitro study revealed that immunosuppressive drugs may impact on T cell cytokine profile, suggesting an inhibition of Th1 response in the presence of a high concentration of tacrolimus, while the impact of mycophenolate seemed to be negligible [24].

Another crucial issue is related to the potential development of anti-HLA antibodies in vaccinated transplanted patients [25]. In our population, we confirmed that also vaccination against SARS-CoV-2 can promote DSA appearance but with no clinical consequences, since none of the patients developed acute renal rejection. On the other hand, Russo et al. reported no DSA appearance after two doses of BNT162b2 vaccine in a kidney transplant population [26]. Even if this aspect remains controversial, DSA monitoring after vaccination might be suggested. 

Regarding the SARS-CoV-2 infection that occurred even after administration of the third dose, no patient required invasive ventilation and ICU admission, and no deaths were observed, confirming that the third dose is protective against severe complications and mortality from COVID-19, as demonstrated in healthy subjects [27,28] and immunocompromised subjects [29,30]. 

## 5. Conclusions

The great strength of our study is the simultaneous evaluation of humoral and cell-mediated responses elicited by BNT162b2 vaccination and the monitoring of anti-HLA antibodies serum levels in a cohort of KTRs during the overall period of vaccination, starting from pre-vaccination to three months after administration of the third dose. All these aspects have been poorly investigated in the literature. On the other hand, our paper has several limitations, including the lack of an age-matched control group, the small sample size of our cohort, the lack of the determination of SARS-CoV-2 variants of concern (VOCs) in infected subjects and the lack of some follow-up time points due to the participation of patients from regions different from that of the hospital. Additionally, the lack of anti-N IgG determination might impact on the results. However, since all breakthrough infections were reported between January and February 2022, when the Omicron variant accounted for >95% of the circulating strains in Italy [31], it is conceivable that the large majority of infections were related to the Omicron variant. Some studies demonstrated an increase of seroconversion after booster doses in transplant recipients, but further studies for the evaluation of long-term immune responses are still ongoing [32,33].

## Figures and Tables

**Figure 1 vaccines-10-00921-f001:**
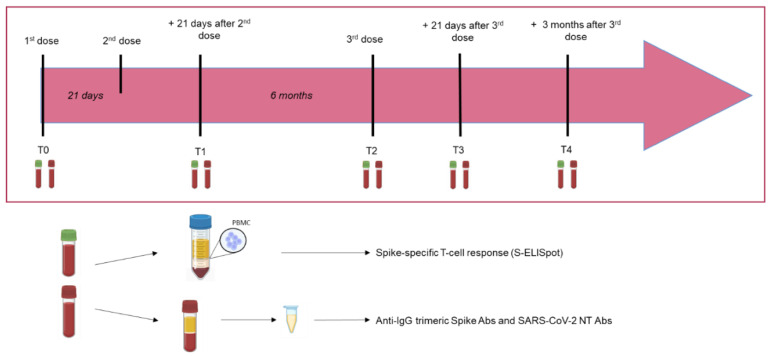
Timeline of the study design and samples’ collection.

**Figure 2 vaccines-10-00921-f002:**
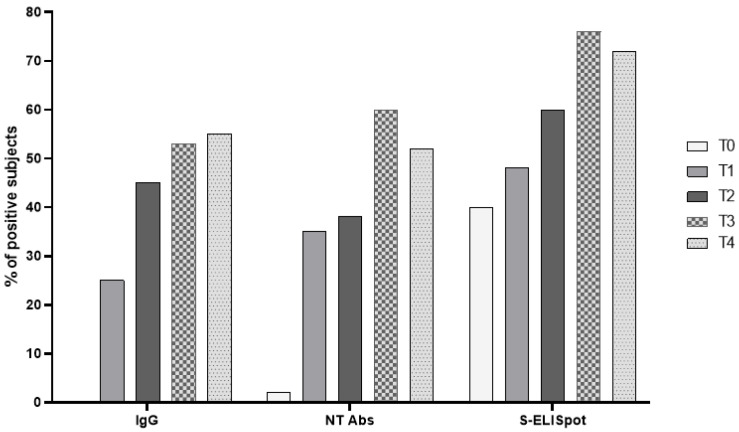
The prevalence of responders for each parameter was determined at T0 (baseline), T1 (three weeks after the second dose), T2 (at the time of the third dose), T3 (three weeks after the third dose) and T4 (three months after the third dose).

**Figure 3 vaccines-10-00921-f003:**
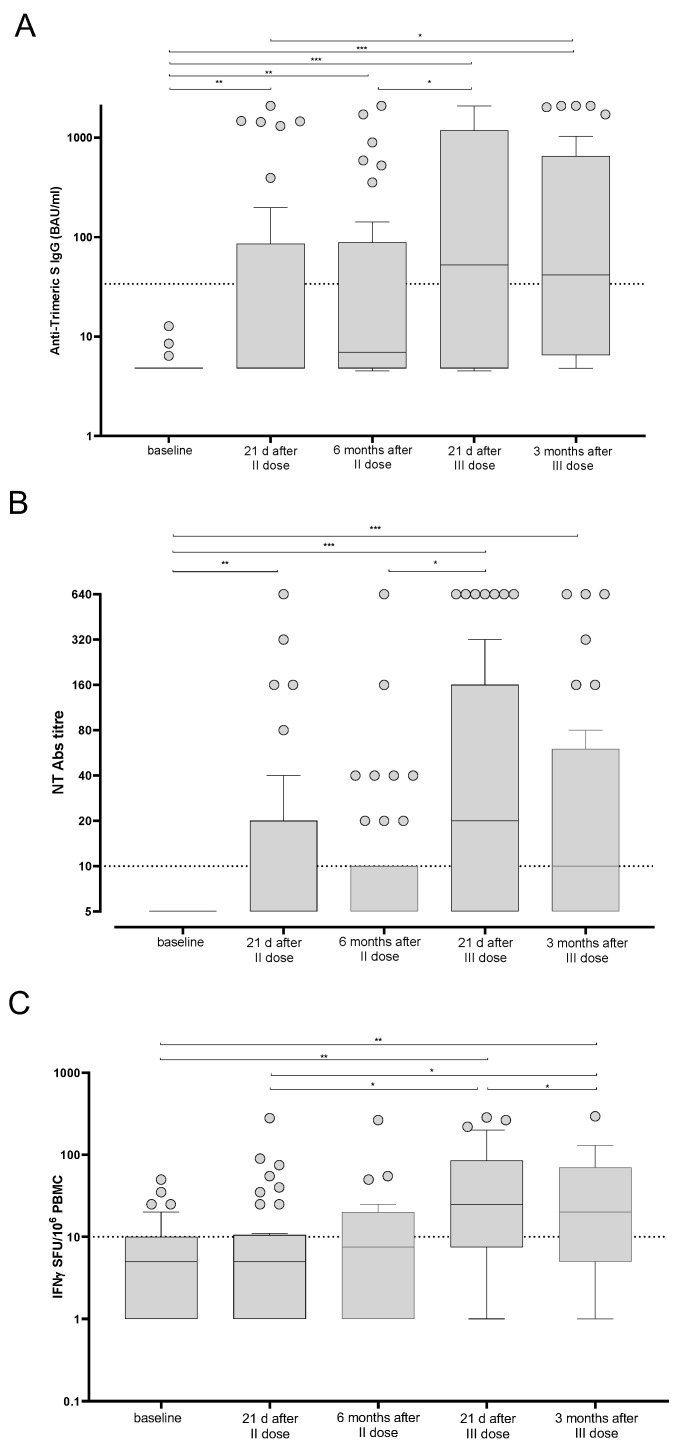
Trimeric anti-Spike IgG (**A**), SARS-CoV-2 NT Abs (**B**) and Spike-specific T cell response (**C**) were measured in BNT162b2-vaccinated kidney transplant recipients at five time points. The results are shown as median and IQR. Significant differences are reported in each graph. (*) *p* < 0.05; (**) *p* < 0.01; (***) *p* < 0.001.

**Figure 4 vaccines-10-00921-f004:**
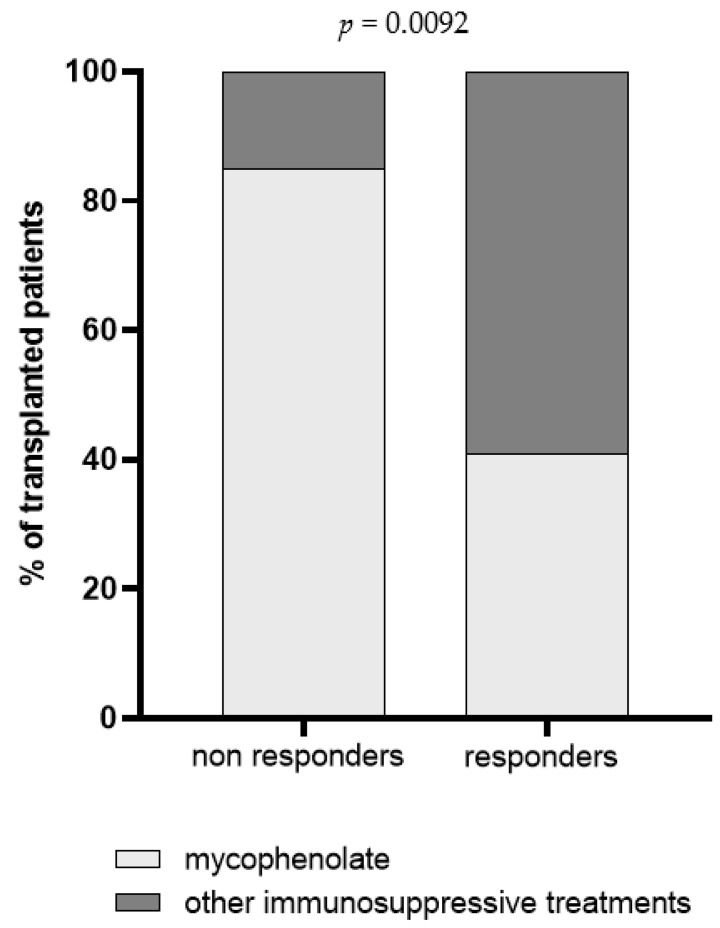
Prevalence of non-responder and responder patients for trimeric IgG three weeks after the third dose. Patients were classified according to immunosuppressive regimen in mycophenolate-treated patients (light grey) and patients treated with a combination of immunosuppressive drugs excluding mycophenolate (dark grey). The *p* value was calculated using Fisher’s exact test.

**Figure 5 vaccines-10-00921-f005:**
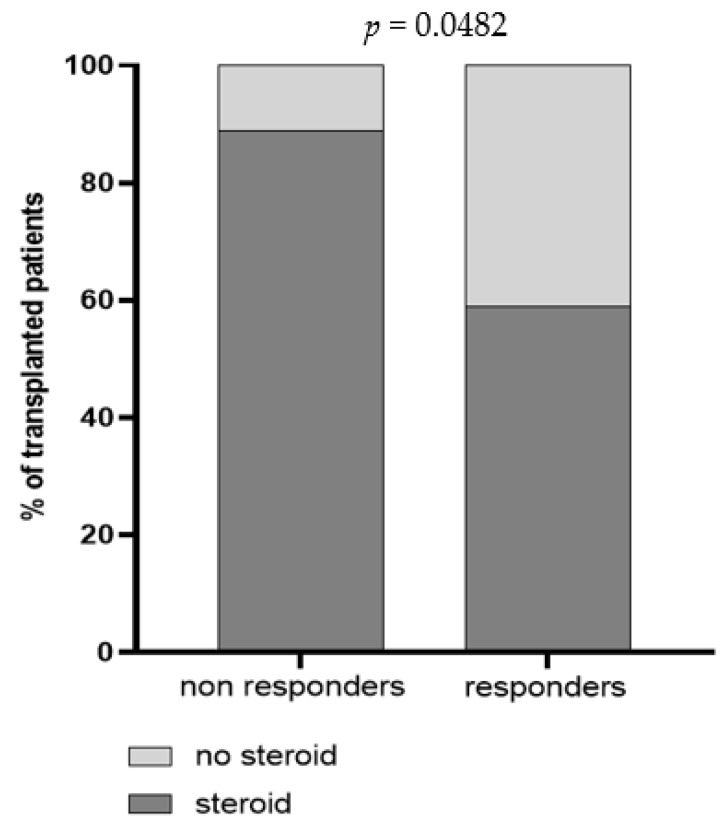
Prevalence of non-responder and responder patients for trimeric SARS-CoV-2 NT Abs three weeks after the third dose. Patients were classified according to steroid administration in steroid-untreated patients (light grey) and steroid-treated patients (dark grey). The *p* value was calculated using Fisher’s exact test.

**Table 1 vaccines-10-00921-t001:** Characteristics of the enrolled KTRs.

Variable	
**Gender, N (%)**	
male	23 (51.1)
female	22 (48.9)
**Age, years**	
mean (IQR)	52.6 (47.2–60)
**Cause of ESRD, N (%)**	
glomerulonephritis	15 (33.3)
hereditary nephropathy	10 (22.2)
diabetes	3 (6.6)
hypertension	9 (20)
urological causes	5 (11.1)
miscellaneous	3 (6.6)
**Transplant age (months)**	
median (IQR)	51 (26–78.5)
**Immunosuppressive treatment, N (%)**	
CNI	42 (93.3)
*- Tacrolimus*	37 (82.2)
*- Cyclosporine*	5 (11.1)
mTORi	14 (31.1)
antimetabolites	30 (66.6)
steroid	32 (71.1)

N: number; %: percentage; ESRD: end-stage renal disease; IQR: interquartile range; CNI: calcineurin inhibitors; mTORi: mammalian target of rapamycin inhibitors.

**Table 2 vaccines-10-00921-t002:** Multiple linear regression analysis of factors potentially associated with the response to three doses of vaccine in transplant recipients.

Dependent Variable	Independent Variable	Estimate β Coefficient	95% Confidence Interval	*p* Value
S Trimeric (Log_10_BAU/mL)	Intercept	3.442	1.774 to 5.110	<0.001
	Age	−0.011	−0.0374 to 0.015	0.401
	Sex (F)	−0.022	−0.669 to 0.624	0.944
	Use of mycophenolate	−0.850	−1.524 to −0.176	0.015
	Use of steroid	−0.653	−1.373 to 0.068	0.075
Nt Abs (Log_10_ titer)	Intercept	3.009	1.709 to 4.310	<0.001
	Age	−0.017	−0.038 to 0.003	0.095
	Sex (F)	−0.049	−0.553 to 0.455	0.845
	Use of mycophenolate	−0.436	−0.962 to 0.089	0.101
	Use of steroid	−0.406	−0.968 to 0.156	0.152
Spike-specific T cells (Log_10_ Spots)	Intercept	3.446	2.246 to 4.646	<0.001
	Age	−0.032	−0.051 to −0.013	0.002
	Sex (F)	−0.088	−0.532 to 0.356	0.689
	Use of mycophenolate	0.118	−0.346 to 0.582	0.609
	Use of steroid	−0.658	−1.184 to −0.132	0.016

**Table 3 vaccines-10-00921-t003:** Immunological parameters, symptoms and therapy of 11 kidney transplant recipients with documented SARS-CoV-2 infection after the third dose of the BNT162b2 vaccine.

Patient ID	Days *	IgG Abs	SARS-CoV-2 NT Abs	S-ELISpot	Symptoms	Therapy
#1	92	neg	1:10	10	none	no
#2	117	125	neg	na	cold hoarseness	molnupiravir
#3	103	113	1:10	15	None	no
#4	106	>2080	1:640	575	cold muscolar pain	remdesivir
#5	100	>2080	1:320	65	cough	no
#6	96	neg	neg	neg	cold	MAbs
#7	117	neg	neg	neg	pneumonia, acute respiratory distress and acute kidney injury,	no
#8	110	neg	neg	10	pneumonia, and acute kidney injury	no
#9	117	neg	1:40	15	ageusia	MAbs
#10	130	183	1:40	na	cough	no
#11	105	neg	neg	25	cold, cough	MAbs

* Days after administration of the third dose; MAbs: Monoclonal antibodies; IgG Abs BAU/mL; S-ELISpot: Spike-specific T cell response; neg: negative result; na: not available.

## Data Availability

Data available upon request to author due to data privacy and ethical restrictions.

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
