# Peer review of "Effect of a Third Dose of SARS-CoV-2 mRNA BNT162b2 Vaccine on Humoral and Cellular Responses and Serum Anti-HLA Antibodies in Kidney Transplant Recipients"

_vaccines, 2022, doi:10.3390/vaccines10060921_

Round 1

Reviewer 1 Report

No further comments

Author Response

Replies to reviewers

Reviewer 3

  1. The possibility of such infections and their impact on the results should however be addressed in limitations of the study which was not done.

The limitations of the study were changed to include this weakness as well

  1. There is an error in the numbering of the references (item 34 is missing), which results from the error in items 16-18.

Thank you for your careful revision. References have been revised and the errors reported have been corrected, item 34 was a mistake.

  1. I am referring again to the issue raised in note 5 in my round-1  review. The text of the discussion still does not take into account other factors that may affect the immune response. The studies that analyzed this issue were cited by authors according my recommendations, but these factors were not mentioned. I propose to list these potential factors in one sentence.

The text has been modified as your suggestion

Academic Editor Notes

Your manuscript certainly is improved, but there are still a number of outstanding issues, as indicated by the reviewer reports. In particular you should pay attention to the following:

The T cell detection method is not specific for SARS-CoV-2, therefore the findings and conclusions should be discussed in that perspective. Please also provide separate results for patients with positive T cell response at time point 0.

Dear Editor, as your comment, we performed a separate analysis, but no differences were observed in the two groups. A supplementary figure 1 has been added and paragraph 3.3 has been amended.

Anti Spike antibody level remained constant during a period of 6 months after the second vaccine dose. This is unusual (as compared to other publications) and this issue should be discussed/explained in the manuscript

Thank you for your comment. This aspect has been specified in paragraph 3.3.

Univariate analysis has been used, but this should be multivariate analysis. Please redo the statistics and ammend the conclusions if needed.  

A multivariate analysis has been performed and paragraph 3.4 has been implemented with these results (new table 2).

Reviewer 2 Report

  • English language editing is needed

Author Response

(The authors gave the same response as above.)

Academic Editor Notes

Your manuscript certainly is improved, but there are still a number of outstanding issues, as indicated by the reviewer reports. In particular you should pay attention to the following:

The T cell detection method is not specific for SARS-CoV-2, therefore the findings and conclusions should be discussed in that perspective. Please also provide separate results for patients with positive T cell response at time point 0.

Dear Editor, as your comment, we performed a separate analysis, but no differences were observed in the two groups. A supplementary figure 1 has been added and paragraph 3.3 has been amended.

Anti Spike antibody level remained constant during a period of 6 months after the second vaccine dose. This is unusual (as compared to other publications) and this issue should be discussed/explained in the manuscript

Thank you for your comment. This aspect has been specified in paragraph 3.3.

Univariate analysis has been used, but this should be multivariate analysis. Please redo the statistics and ammend the conclusions if needed.  

A multivariate analysis has been performed and paragraph 3.4 has been implemented with these results (new table 2).

Reviewer 3 Report

  1. I refer to my Note # 1 from my round-1 review. From the authors' responses, I understand that anti-nucleocapsid antibody determinations were not performed to monitor potential asymptomatic infections. The possibility of such infections and their impact on the results should however be addressed in limitations of the study which was not done.
  2. There is an error in the numbering of the references (item 34 is missing), which results from the error in items 16-18.
  3. I am referring again to the issue raised in note 5 in my round-1  review. The text of the discussion still does not take into account other factors that may affect the immune response. The studies that analyzed this issue were cited by authors according my recommendations, but these factors were not mentioned. I propose to list these potential factors in one sentence.

Author Response

(The authors gave the same response as above.)

Academic Editor Notes

Your manuscript certainly is improved, but there are still a number of outstanding issues, as indicated by the reviewer reports. In particular you should pay attention to the following:

The T cell detection method is not specific for SARS-CoV-2, therefore the findings and conclusions should be discussed in that perspective. Please also provide separate results for patients with positive T cell response at time point 0.

Dear Editor, as your comment, we performed a separate analysis, but no differences were observed in the two groups. A supplementary figure 1 has been added and paragraph 3.3 has been amended.

Anti Spike antibody level remained constant during a period of 6 months after the second vaccine dose. This is unusual (as compared to other publications) and this issue should be discussed/explained in the manuscript

Thank you for your comment. This aspect has been specified in paragraph 3.3.

Univariate analysis has been used, but this should be multivariate analysis. Please redo the statistics and ammend the conclusions if needed.  

A multivariate analysis has been performed and paragraph 3.4 has been implemented with these results (new table 2).

This manuscript is a resubmission of an earlier submission. The following is a list of the peer review reports and author responses from that submission.

Round 1

Reviewer 1 Report

In this paper, the authors have analyzed both serological and cellular response to the vaccination by BNT162B2 in 47 patients with a renal transplantation. They have analysed the immune response after each vaccination, and conclude that the 3rd dose is associated with the highest cellular response.

Major comments:

- In the discussion, it is indicated that some patients were infected by the Omicron variant. Therefore, the period when the study was realized shuld be indicated in patients and samples, as well as the different types of variants during the study.

  • The authors show that after the third dose, 75,6 % of the patients had a cellular response, however, before vaccination, almost 40 % of the patients had a positive cellular test. The authors suggest that it is due to a cross reaction with other common coronaviruses. Therefore, in my opinion, the vaccination induced a cellular response only in 30% of the patients. In contrast, a serological response was observed in 55% of patients. This should considered in a revised version. Lastly it is not actually known which type of immunological response in the more efficient. Therefore, it should be interesting  to compare the frequency of patients who have only one type of response (serological or cellular) and both responses, in patients who experienced a COVID-19 and in patients who were protected against the disease.
  • One patients was considered as free of disease at the time of enrollment because he had a negative serological assay, but had neutralyzing antibodies. Dis the authors tested this patients with another kit than the Diasorin ? Is it a false negative of the serology or a false positive of the neutralysing assay?
  • It is indicated in "patients and methods" than only COVID-19 naive patients were included, but in paragraph 3.6, line 237, it is indicated that 2 patients reported a previous COVID-19 before vaccination. This should be explaines.

Minor comments:

47 patients were included in this study. However, in paragraph 3.2, "immunogenicity of BNT..." it is indicated that eLISPOT was tested in 42 patients. This limitation should be indicated in the Methods section. Similarly, in the figure 3, only dots for 28 patients are shown in the figure C, 3 months after the third dose. Is it a mistake or only 28 patients were tested ?

Author Response

Reviewer 1

Comments and Suggestions for Authors

In this paper, the authors have analyzed both serological and cellular response to the vaccination by BNT162B2 in 47 patients with a renal transplantation. They have analysed the immune response after each vaccination, and conclude that the 3rd dose is associated with the highest cellular response.

Major comments:

- In the discussion, it is indicated that some patients were infected by the Omicron variant. Therefore, the period when the study was realized should be indicated in patients and samples, as well as the different types of variants during the study.

Response: The period of enrollment and follow-up was included in material and methods section (changes are highlighted in red in the text, page 2, line 81 and line 85) . We pointed out that variants were not studied in this paper but, according to the epidemiological situation in Italy, it is conceivable that positive subjects were infected with Omicron variant.

- The authors show that after the third dose, 75,6 % of the patients had a cellular response, however, before vaccination, almost 40 % of the patients had a positive cellular test. The authors suggest that it is due to a cross reaction with other common coronaviruses. Therefore, in my opinion, the vaccination induced a cellular response only in 30% of the patients. In contrast, a serological response was observed in 55% of patients. This should considered in a revised version. Lastly, it is not actually known which type of immunological response in the more efficient. Therefore, it should be interesting to compare the frequency of patients who have only one type of response (serological or cellular) and both responses, in patients who experienced a COVID-19 and in patients who were protected against the disease.

Response: This concept was underlined in the discussion section (page 9, line 264-269). However, we further underlined that median T-cell response increased over time, thus suggesting that vaccination is able to increase cell mediated response also in those subjects showing a pre-existing immune response. Due to the low number of subjects included in the study, no significant differences were appreciated.

-One patient was considered as free of disease at the time of enrollment because he had a negative serological assay, but had neutralyzing antibodies. Did the authors tested this patients with another kit than the Diasorin ? Is it a false negative of the serology or a false positive of the neutralysing assay?

Response: No further tests have been made. Considering the history of the patients, it is conceivable that in two patients there was a false positive of the neutralizing assay. Thus, they have been eliminated from the analyses and all the results were reviewed.

- It is indicated in "patients and methods" than only COVID-19 naive patients were included, but in paragraph 3.6, line 237, it is indicated that 2 patients reported a previous COVID-19 before vaccination. This should be explaines.

Response: We specified that further analyses were performed only in COVID-19 naïve patients at baseline, those two patients were excluded from the analyses, line 87 page 2.

Minor comments:

47 patients were included in this study. However, in paragraph 3.2, "immunogenicity of BNT..." it is indicated that ELISPOT was tested in 42 patients. This limitation should be indicated in the Methods section. Similarly, in the figure 3, only dots for 28 patients are shown in the figure C, 3 months after the third dose. Is it a mistake or only 28 patients were tested?

Response: Some drop out are present in the study, due to the prospective nature of the study. Thus, we included the missing time points as limitation of the study (page 10 line 308-309).

Reviewer 2 Report

  1. English language editing is required
  2. Introduction
    1. the background is not clearly explained (lines 40 and forward) .
    2. you use expressions like "dramatically increased' , and "most" , if you can provide numbers / stats from the cited references.
  3. Methods
    1. could you provide dates of the study conduction
    2. if you could provide flow chart for patients' recruitment (screened, included, excluded)
  4. results
    1. immunosupression: you present in table 1 % of each immunosupression class. most patients are on triple regimn as you state in the text. you may consider to add the most common combinations to the table .
    2. 40% of your patients had a positive T cell Elispot response at t0.
      1. it is not completely clear that these patients did not have COVID-19 infection .
      2. it would be intresting to have a separate analysis of this group regarding type and titers of their immune response
  5. Discussion 
    1. many publications report a decrease in antibody level 6 months after vaccination,in your results IgG and NT antibody leves increased   at that time point (T2) how do you explain your results.
    2. regarding the association of immune response and different parameters
      1. antimetabolite immunosupression - was this association independent or confounders are possible explanation (like being usually the third class added to the regimen. )
      2. association with age was demonstrated in univariate analysis , did you perform any adjustment / multivariate analysis?
    3. patients with brekthrough infection: how is their immune response compare with patients who did not ave breakthrough infection?
  6. Conclusions it is advised to formulate your conclusions.

Author Response

Reviewer 2

Comments and Suggestions for Authors

  1. English languageediting is required

Response: the paper has been revised as suggested

  1. Introduction
    1. the background is not clearly explained (lines 40 and forward) .

Response: the text has been amended as your suggestion

    1. you use expressions like "dramatically increased' , and "most" , if you can provide numbers / stats from the cited references.

Response: the text has been revised as your advice

  1. Methods
    1. could you provide dates of the study conduction

Response: Period of the study was provided in Material and Methods section (page 2 line 81 and 85)

    1. if you could provide flow chart for patients' recruitment (screened, included, excluded)

Response: we revised analyses and results including only naïve patients

  1. Results
    1. immunosuppression: you present in table 1 % of each immunosupression class. most patients are on triple regimen as you state in the text. you may consider to add the most common combinations to the table.

Response: results paragraph 3.1 and table 1 have been modified as your suggestion

  1. 40% of your patients had a positive T cell Elispot response at t0. it is not completely clear that these patients did not have COVID-19 infection

Response: patients were considered “COVID-19 naïve” since they did not reported positive SARS-CoV-2 RNA. Patients with reported T-cell response measured by ELISpot assay might be previously exposed to other common coronaviruses as largely demonstrated in our previous paper and in other studies. (line 87 page 2)

  1. it would be intresting to have a separate analysis of this group regarding type and titers of their immune response

Response: We agreed with the reviewer comment; however, due to the low number of subjects included in the study no differences might be appreciated.

  1. Discussion
    1. Many publications report a decrease in antibody level 6 months after vaccination, in your results IgG and NT antibody levels increased at that time point (T2) how do you explain your results regarding the association of immune response and different parameters.

Response: It is conceivable that, due to the immunosuppressed status of these patients the development of immune response elicited by vaccination might be delayed than respect to that observed in healthy subjects. However, the small sample subset of patients included in this study might lead to a misinterpretation of the results. Another hypothesis might be related to possible SARS-CoV-2 infections (mainly asymptomatic) that might have been occurred between the second dose administration and time point T3.

  1. antimetabolite immunosupression - was this association independent or confounders are possible explanation (like being usually the third class added to the regimen.

Response: Many papers demonstrated that MMF had a highly significant effect on the development of antibodies after vaccination in transplanted patients (Kantauskaite M et al, Am J Transpl 2022) as in patients with immune mediated diseases such as rheumatic diseases.  The American College of Rheumatology recently recommended withholding mycophenolate for 1 week after vaccination to enhance immunogenicity in these patients.  (Connolly CM et al, Ann Rheum Dis February 2022)

  1. association with age was demonstrated in univariate analysis, did you perform any adjustment / multivariate analysis?

Response: Multivariate analysis was not performed due to the low sample size that limits the statistical power but the distribution of double and triple immunosuppression regimen was similar in the two groups.

    1. patients with breakthrough infection: how is their immune response compare with patients who did not Have breakthrough infection?

Response: No difference were reported probably due to the low number of patients included in the study. However, we previously reported in a larger cohort of healthcare subjects that no differences were observed in terms of immune response (both humoral and cell-mediated responses), thus suggesting that these factors might be not related to the increased risk to be infected (Rovida et al. IJID 2021).

  1. Conclusions it is advised to formulate your conclusions.

Response: A conclusive section was added at the end of the manuscript (page 10 lines 303-315)

Reviewer 3 Report

  1. SARS-CoV-2 infection could have occurred not only before the enrollement  in the study, but also during the study. It could have influenced the tested results of the humoral and cellular responses. Such an infection could have been asymptomatic, unnoticed by the patient. Has the possibility of such a situation been ruled out by testing, for example, the titer of anti-nucleocapside antibodies?

2. In the opinion of the reviewer, the antibody titer should be analyzed only in seroconverted patients. Otherwise, their titer is greatly underestimated and it is difficult to compare the results to other studies

3. The authors report that only patients without prior SARS-CoV-2 infection (line 80) were enrolled in the study. At the same time they contradict it in the results section (line 237).

4. The authors mention the very severe course of COVID-19 among hemodialysis patients and kidney transplant recipients without citing any literature in this regard (line 247). I propose to quote eg .study of Puchalska et al showing the fatality rate due to COVID-19 in HD patients up to 43% in the oldest HD individuals: Pol Arch Intern Med. 2021 Jul 30;131(7-8):643-648. doi: 10.20452/pamw.16028) and study of Azzi et al. (Transplantation 2021105, 37–55. doi: 10.1097/TP.0000000000003523) on kidney transplant recipients.

5. Factors that may affect the immune response after vaccination, and which were not mentioned by the authors in the discussion, include among others: the amount of immunosuppressive drugs taken, vintage transplant and the type of vaccine, i.e. - better response to mRNA-1273 (eg. Debska-Slizien et al: Vaccines doi: 10.3390/vaccines9101165). Moreover, some studies indicate that subsequent booster doses (third and fourth) significantly increase the percentage of kidney transplant recipients who seroconvert (3rd dose: Tylicki et al: Vaccines: doi: 10.3390/vaccines10010056  and 4th dose: Caillard et al: Annals of Internal Medicine: doi.org/10.7326/L21-0598). It is worth mentioning in the discussion.

6. In the abstract and discussion, the authors should emphasize the great advantage of this work, which is the complexity of analyzes including humoral and cellular responses as well as HLA antibodies appereance and rejection episodes

7. Including conclusion remarks at the end of the discussion and the abstract should  be considered

Author Response

Reviewer 3

Comments and Suggestions for Authors

  1. SARS-CoV-2 infection could have occurred not only before the enrollement  in the study, but also during the study. It could have influenced the tested results of the humoral and cellular responses. Such an infection could have been asymptomatic, unnoticed by the patient. Has the possibility of

 such a situation been ruled out by testing, for example, the titer of anti-nucleocapside antibodies?

Response: we agree with the reviewer but to perform this additional test as requested by the reviewer we need some more time (at least two weeks) to perform and analyze in the collected serum samples

  1. In the opinion of the reviewer, the antibody titer should be analyzed only in seroconverted patients. Otherwise, their titer is greatly underestimated and it is difficult to compare the results to other studies

Response: analyses carried out were consistent with the design and study protocol approved by EC.

  1. The authors report that only patients without prior SARS-CoV-2 infection (line 80) were enrolled in the study. At the same time, they contradict it in the results section (line 237).

Response: Thanks for your comment. The studied population has been revised, we analyzed only naïve patients.

  1. The authors mention the very severe course of COVID-19 among hemodialysis patients and kidney transplant recipients without citing any literature in this regard (line 247). I propose to quote eg .study of Puchalska et al showing the fatality rate due to COVID-19 in HD patients up to 43% in the oldest HD individuals: Pol Arch Intern Med. 2021 Jul 30;131(7-8):643-648. doi: 10.20452/pamw.16028) and study of Azzi et al. (Transplantation2021, 105, 37–55. doi: 10.1097/TP.0000000000003523) on kidney transplant recipients.

Response: Thanks for your comment, we added the references as your suggestion

5.Factors that may affect the immune response after vaccination, and which were not mentioned by the authors in the discussion, include among others: the amount of immunosuppressive drugs taken, vintage transplant and the type of vaccine, i.e. - better response to mRNA-1273 (eg. Debska-Slizien et al: Vaccines doi: 10.3390/vaccines9101165). Moreover, some studies indicate that subsequent booster doses (third and fourth) significantly increase the percentage of kidney transplant recipients who seroconvert (3rd dose: Tylicki et al: Vaccines: doi: 10.3390/vaccines10010056  and 4th dose: Caillard et al: Annals of Internal Medicine: doi.org/10.7326/L21-0598). It is worth mentioning in the discussion.

Response: Thanks for your comment, we added the references as your suggestion

7.In the abstract and discussion, the authors should emphasize the great advantage of this work, which is the complexity of analyzes including humoral and cellular responses as well as HLA antibodies appereance and rejection episodes

Response: Thanks for your comment, the text has been amended as your suggestion

  1. Including conclusion remarks at the end of the discussion and the abstract should be considered

Response: Discussion and abstract have been modified as your suggestion